# Exceptionally High 2018 Equilibrium Line Altitude on Taku Glacier, Alaska

Mauri Pelto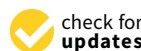

Department of Environmental Sciences, Nichols College, Dudley, MA 01571, USA; mspelto@nichols.edu;
Tel.: +508-213-2168

**Abstract:** The Juneau Icefield Research Program (JIRP) has been examining the glaciers of the Juneau Icefield since 1946. The height of the transient snowline (TSL) at the end of the summer represents the annual equilibrium line altitude (ELA) for the glacier, where ablation equals accumulation. On Taku Glacier the ELA has been observed annually from 1946 to 2018. Since 1998 multiple annual observations of the TSL in satellite imagery identify both the migration rate of the TSL and ELA. The mean ELA has risen 85 ± 10 m from the 1946–1985 period to the 1986–2018 period. In 2018 the TSL was observed at: 900 m on 5 July; 975 m on 21 July; 1075 m on 30 July; 1400 m on 16 September; and 1425 m on 1 October. This is the first time since 1946 that the TSL has reached or exceeded 1250 m on Taku Glacier. The 500 m TSL rise from 5 July to 30 July, 8.0. $md^{-1}$, is the fastest rate of rise observed. This combined with the observed balance gradient in this region yields an ablation rate of 40–43 $mmd^{-1}$, nearly double the average ablation rate. On 22 July a snow pit was completed at 1405 m with 0.93 m w.e. (water equivalent), that subsequently lost all snow cover, prior to 16 September. This is one of eight snow pits completed in July providing field data to verify the ablation rate. The result of the record ELA and rapid ablation is the largest negative annual balance of Taku Glacier since records began in 1946.

**Keywords:** glacier runoff; transient snow line; equilibrium line altitude; climate change; Landsat; glacier ablation

---

## 1. Introduction

The transient snowline (TSL) is the location of the transition from snow cover to bare glacier ice or older firn on a glacier at any time during a hydrologic year [1]. The equilibrium line altitude (ELA) is the altitude of the TSL at the end of the ablation season, its highest elevation during the hydrologic year. The TSL can be identified through GPS field mapping, aerial photography or satellite imagery [2–4]. At the TSL, the glacier mass balance for the date of observation is zero for temperate maritime glaciers, as they lack superimposed ice. Superimposed ice develops when percolating meltwater refreezes on a cold underlying glacier surface during summer; on temperate glaciers a cold enough surface does not exist in summer. TSL becomes visible when all snow has melted at a point on the glacier [5]. The migration of the TSL up-glacier through the remainder of the melt season can be used in glacier mass balance determination, glacier runoff calculation and as a model input or validation tool [3,4,6]. In standard glaciological mass balance field assessment, ablation has been measured by drilling stakes into a glacier and observing the change in the height of the stake above the glacier surface [7]. Ablation is the product of the change in the height of the stake above the glacier surface and the density of snow or ice. Migration of the TSL can provide an additional measure of ablation if the balance gradient in the elevation range that the TSL traverses is known [3,4,8]. Snowpack ablation at the TSL is the product of the balance gradient and TSL migration rate [3]. Hulth et al. [8] used repeat snowline

mapping to determine accumulation, based on the observed and modeled amount of melt from TSL movement; this is simply the inverse of determining snowpack ablation rate.

The current availability of satellite imagery from many sources ensures sufficient images for many glaciers in the ablation season for most years since early 1990s. Because of this ease of observation of the TSL, repeat imaging during a single season has been used on Chhota Shigri Glacier, India; Abramov Glacier, Kyrgyzstan; Storglaciaren, Sweden; and Vernagtferner, Austria to aid in mass balance assessment [5,6,8,9].

Taku Glacier is the site of ongoing annual mass balance and ELA observations since 1946 [10,11]. All mass balance data is submitted to the World Glacier Monitoring Service (WGMS) [12]. In this study we focus only on the ELA observation record. The mass balance record has been validated with independent elevation change based assessments [11]. Observations of TSL migration are used to determine ablation on the Taku Glacier of the Juneau Icefield, Alaska, where by July snowpack density is consistent across the Juneau Icefield [10,11,13]. This method does not provide a measure of ablation for other regions of the glacier, such as the lower ablation zone where the balance gradient is not measured annually or in the upper accumulation zone that the TSL does not transect. Taku Glacier has a strong maritime climate and has experienced an increase in ablation season temperature that has driven recession of all other glaciers of the Juneau Icefield. Taku Glacier experienced a period of positive mass balance with a mean average balance of $+0.40$ ma$^{-1}$ from 1946 to 1985. From 1986 to 2018 the warming has led to a mean annual balance of $-0.19$ ma$^{-1}$ [11,12]. This substantial shift has ended the period of glacier surface elevation increase [11].

## 2. Materials and Methods

On Taku Glacier probing of the thickness of the annual snow layer was completed in early July of 1984, 1998, 2005, 2014 and 2015 at 100 m intervals along a transect from 950 m to 1200 m identifying the balance gradient in this elevation range, quantifying the annual variability and the small-scale spatial variability [3,11]. This elevation range was selected in 1984 because it encompasses a region where annual snow pits are completed and where the TSL typically reaches its maximum elevation. Snow pits are completed in mid-July annually at the same locations on the Taku Glacier (Figure 1) by the Juneau Icefield Research Program (JIRP) using a consistent method [11]. Some of the snow pits are located in the region transected by the observed rise of the TSL. The depth and density of the annual snow layer are directly measured, providing a measure of the retained snow water equivalent on the day of the measurements. In 2018, 14 snow pits were completed on Taku Glacier; eight were in a region that the TSL subsequently rose above.

Landsat imagery was reviewed and obtained using the EarthExplorer platform [14]. All satellite images used were Level 1 terrain corrected Landsat products. Taku Glacier is in path/row 58/19 and 57/19. The TSL is manually digitized for each scene. The images are Tier 1 images, which contain the highest quality Level-1 Precision Terrain (L1TP) data considered suitable for time-series analysis. The geo-registration is consistent and within prescribed tolerances for Landsat 5 TM, Landsat 7 ETM + and Landsat 8 OLI/TIRS products with a <12 m root mean square error (RMSE) [14]. L1TP data are radiometrically calibrated and orthorectified using ground control points and a Digital Elevation Model DEM to correct for elevation displacement. Ground control points used for L1TP correction are derived from the Global Land Survey 2000 (GLS2000) data set [14]. Elevation determination is completed using the Advanced National Map viewer 3DEP Elevation Index product [15]. The image spatial resolution of 30 m and the registration error of 12 m, combined with mean surface slopes of 0.04–0.08 m m$^{-1}$, yield an error of $\pm$1–2 m in TSL elevation, with a mean of 1.2 m. We report the ELA or TSL to the nearest 5 m as it is an average elevation of a number of points on the glacier surface.

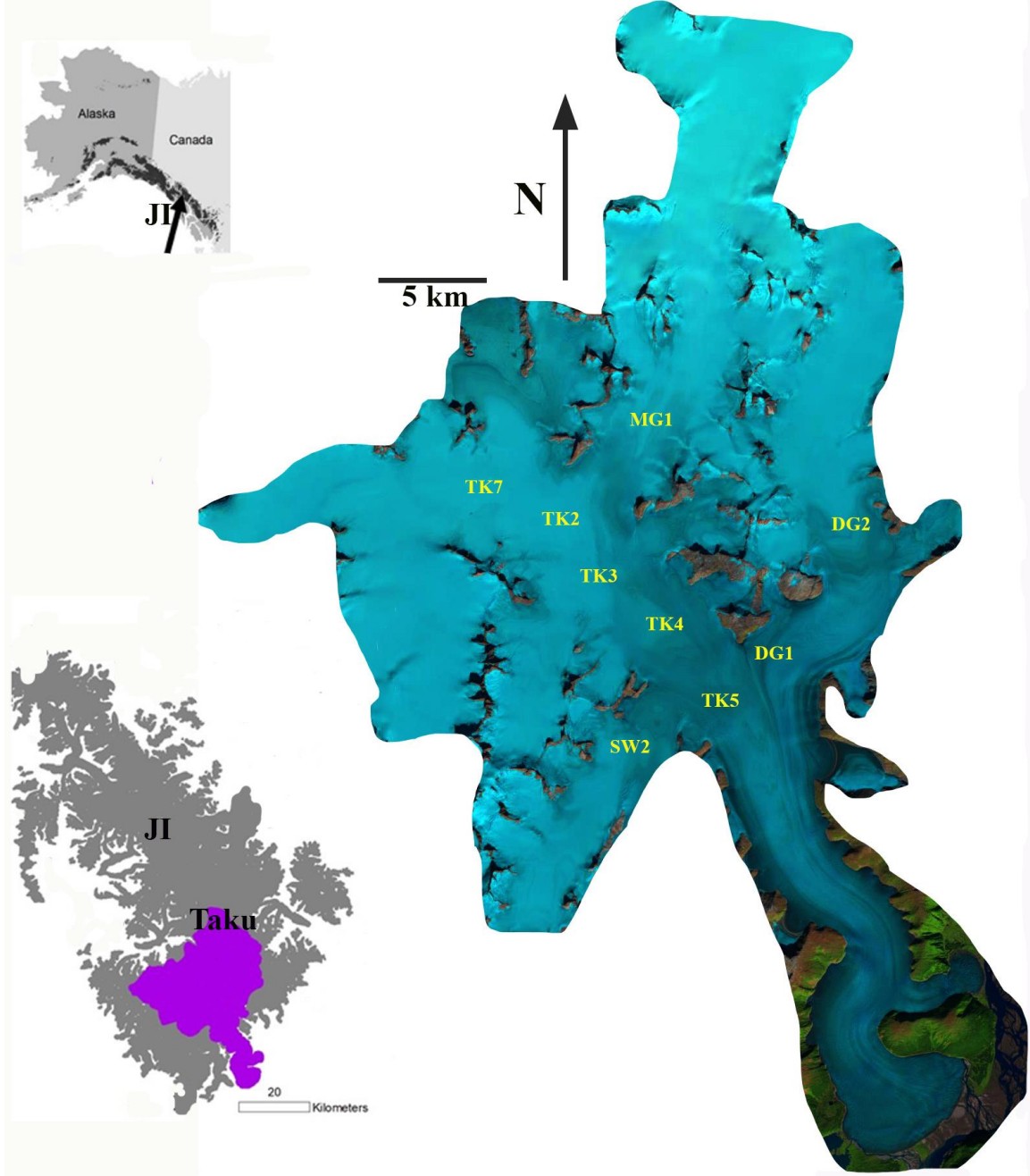

**Figure 1.** Taku Glacier Alaska indicating location of snow pits on a Landsat image from 16 September 2018. Inset indicate location of Taku Glacier on the Juneau Icefield (JI) and the location of the Juneau Icefield in Alaska.

Seventy-three Landsat scenes were identified of sufficient quality for the TSL to be readily identifiable for Taku Glacier from June to early October when the TSL was above 500 m in 1984 and from 1998 to 2018. A probing transect in 1984 coincided with the TSL observations, warranting inclusion of that year. Pairs of scenes that were obtained more than 15 days apart were utilized to determine the rate of TSL rise in the elevation range of the glacier where mass balance observations also are made at 800–1200 m; there are 26 such pairs identified. Shorter time periods lead to higher errors in migration rate; the TSL determination elevation error of 5 m over the minimum time period of 15 days thus yields a migration rate error of 0.33 md$^{-1}$ [4]. In 2018, there were 5 dates with imagery available that allowed TSL identification.

## 3. Results

Observations of the TSL using Landsat images are noted below. Of the 44 observations there are 26 pairs of dates where the TSL is in the elevation reach where field observations have established the balance gradient (Table 1). The annual ELA observations prior to the satellite era were singular observations near the end of the ablation season [10]. All ELA data is reported annually to the WGMS. The TSL observed on 16 September 2018 (month/day/year) and 10 October 2018 are the highest observed on Taku Glacier since annual observations of the ELA began in 1946. The mean ELA from 915 m for the 1946–1985 period to 1000 m for the 1986–2018 period represents a rise of 85 ± 10 m. The time period choice reflects the time of transition from persistent positive mass balance to a regime of net negative mass balance. The previous highest TSL observed, and the only one surpassing 1200 m, was 1225 m in 1997. The vertical difference of 200 m and the horizontal distance between 1225 m and 1425 m on Taku Glacier of 9 km, both indicate the exceptional nature of the observed record ELA in 2018 (Figures 2 and 3).

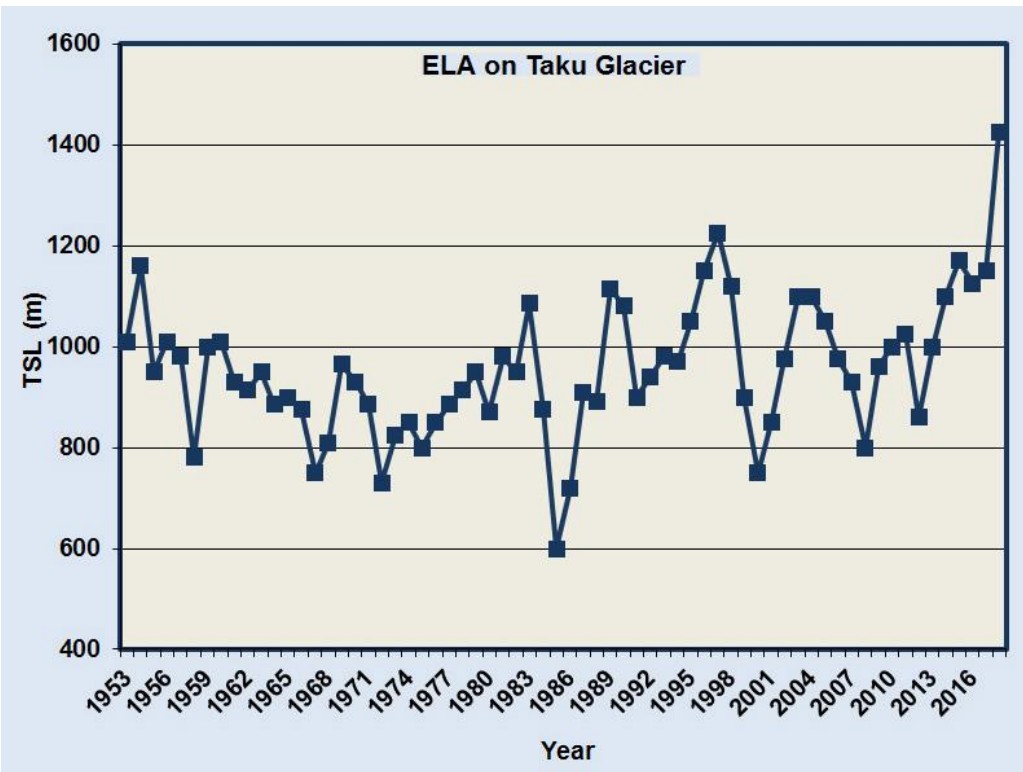

**Figure 2.** Equilibrium line altitude (ELA) observed on Taku Glacier annually since 1946.

The ELA for 2018 is compared to that of 2013, 2014, 2015 and 2017 in Figure 4. The 30 July 2018 image shows approximately the same ELA as in 2015 and 2017.

Prior to 2018 the mean TSL migration rate was 4.34 md$^{-1}$ with a maximum of 6.41 md$^{-1}$ (Figure 5). In 2018 there were three time periods during the ablation season with migration rates ranging from 6.77 to 8.33 md$^{-1}$. The lower migration rate is for the 16 September image when the TSL also exceeded the probing transect gradient range. Each of the three periods represents the most rapid migration rate in the period of satellite record from 1984 to 2018.

Field observation of the balance gradient from probing has fallen within a narrow range during each observation period of 4.7 to 5.7 mm w.e. $m^{-1}$ in the 950–1200 m elevation range with a mean of 5.1 mm w.e. $m^{-1}$. In 2018 the TSL exceeded the top elevation of the probing transect. The observed balance gradient from 1200–1400 m from annual snow pit data for the 1998–2018 period was 2.3 mm w.e. $m^{-1}$.

Given that the balance gradient has been observed to be consistent from year to year [3,11], the product of the mean observed balance gradient and the migration rate yields the ablation rate. The TSL-derived ablation rate is specific to the altitude range over which TSL moved. Prior to 2018, of the 22 periods snowpack ablation ranged from 12 mmd$^{-1}$ to 33 mmd$^{-1}$ with a mean of 23 mmd$^{-1}$ Table 2. The most rapid migration in July 2018 yielded the highest observed snowpack ablation rates of 40 mmd$^{-1}$ and 43 mmd$^{-1}$.

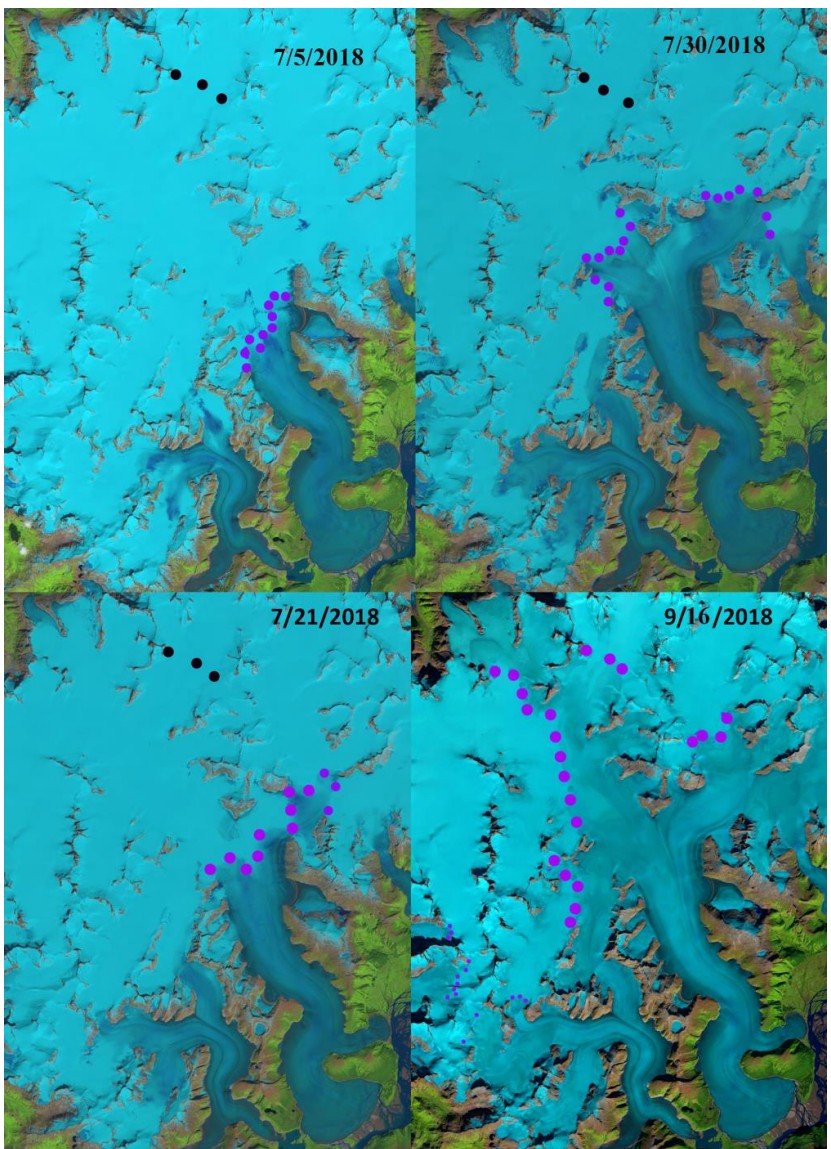

**Figure 3.** The transient snow line on Taku Glacier in Landsat imagery from 2018. TSL on that date is indicated by purple dots, while black dots indicate the 16 September 2018 TSL on the main branch of the glacier.

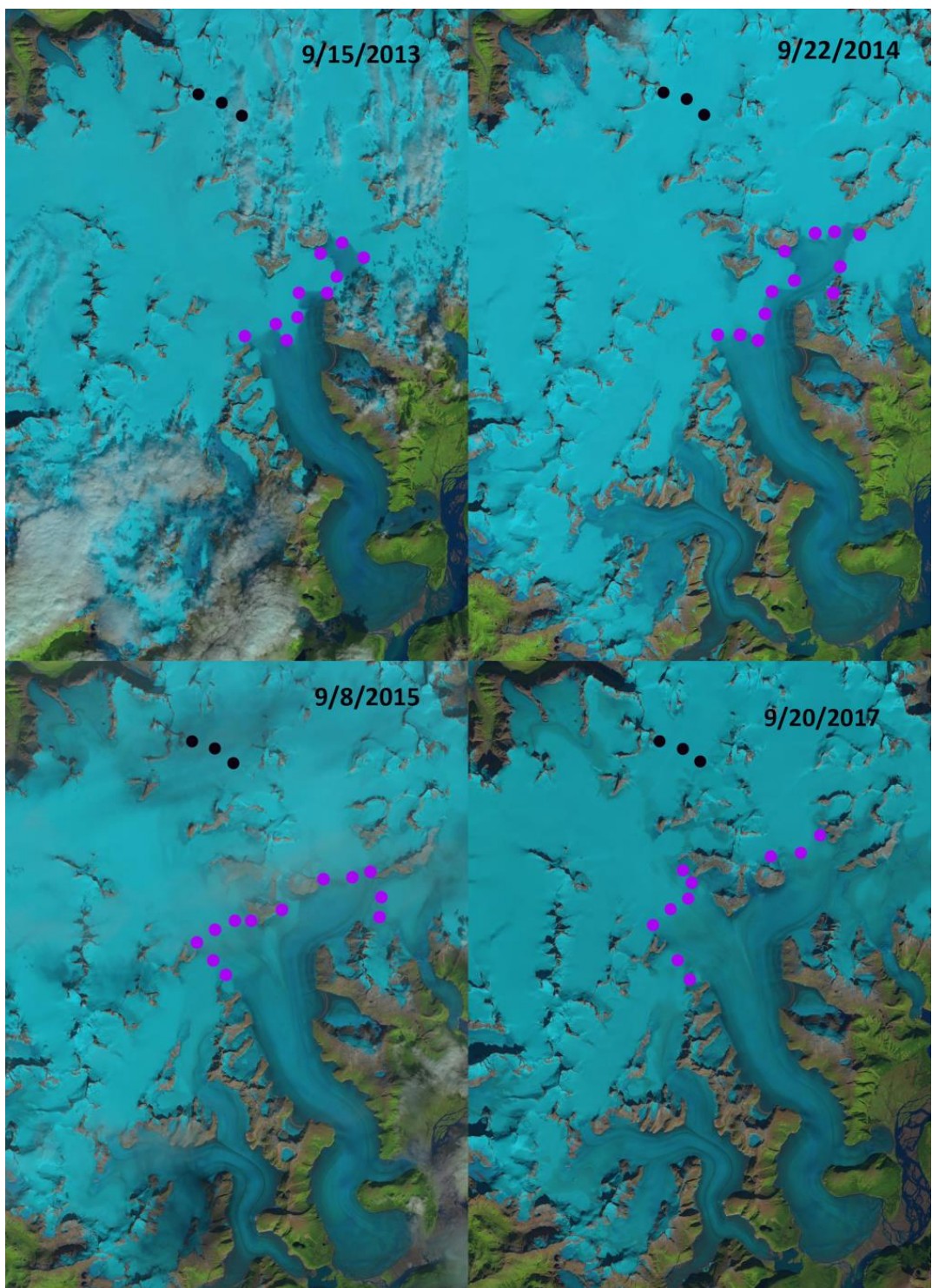

**Figure 4.** The highest transient snow line, ELA, for Taku Glacier in 2013, 2014, 2015 and 2017.

**Table 1.** Transient snow line observation in Landsat images including the migration rate between pairs of images.

| Landsat Image | TSL (m) | Rise (m/day) | Date |
|:---|:---:|:---:|:---:|
| LT50580191984205PAC00 | 800 | | 23 July 1984 |
| LT50580191984232PAC00 | 915 | 4.26 | 18 August 1984 |
| LT50580191984238PAC00 | 975 | 5.30 | 25 September 1984 |
| LT50580191998211PAC00 | 900 | | 30 July 1998 |
| LT50570191998259PAC00 | 1075 | 3.65 | 16 September 1998 |
| LT50580192003193PAC00 | 900 | | 12 July 2003 |
| LE0580192003217EDC02 | 975 | 3.13 | 5 August 2003 |
| LT50570192003234PAC00 | 1075 | 5.88 | 22 August 2003 |
| LT50580192004196PAC00 | 850 | | 14 July 2004 |
| LE0580192004220EDC02 | 930 | 3.33 | 8 August 2004 |
| LT50580192004228PAC00 | 975 | 3.91 | 16 August 2004 |
| LE0580192004236EDC02 | 1000 | 4.38 | 24 August 2004 |
| LT50580192004244PAC00 | 1050 | 5.00 | 31 August 2004 |
| LE0580192005223EDC00 | 925 | | 11 August 2005 |
| LE0580192005254EDC00 | 1000 | 2.42 | 11 September 2005 |
| LE0580192006229EDC00 | 760 | | 29 July 2006 |
| LE0580192006257EDC00 | 975 | 4.48 | 15 September 2006 |
| LE0570192007196EDC00 | 725 | | 15 July 2007 |
| LE0570192007221EDC00 | 850 | 5.21 | 8 August 2007 |
| LE0580192007228EDC00 | 900 | 5.47 | 16 August 2007 |
| LT50570192007245PAC00 | 965 | 4.79 | 2 September 2007 |
| LT50570192007261PAC00 | 975 | 3.13 | 18 September 2007 |
| LE0570192009194EDC00 | 700 | | 13 July 2009 |
| LE0570192009210EDC00 | 800 | 6.25 | 29 July 2009 |
| LE0580192009217SGS02 | 825 | 5.43 | 5 August 2009 |
| LT50580192009257PAC00 | 950 | 3.75 | 14 September 2009 |
| LT50570192011192PAC01 | 800 | | 20 July 2011 |
| LE0580192011255EDC00 | 1000 | 3.70 | 12 September 2011 |
| LC80570192013194LGN00 | 750 | | 13 July 2013 |
| LC80570192013213LGN00 | 850 | 5.26 | 1 August 2013 |
| LC80570192013245LGN00 | 975 | 3.91 | 2 September 2013 |
| LE0580192014199ASN00 | 875 | | 18 July 2014 |
| LC80570192014232LGN00 | 1000 | 3.79 | 20 August 2014 |
| LC80570192015187LGN00 | 850 | | 6 July 2015 |
| LC80580192015226LGN00 | 1100 | 6.41 | 14 August 2015 |
| L1C_T08VNK_A010464_20170623T | 725 | | 23 June 2017 |
| LC08_L1TP_0580192017920-T1 | 1100 | 3.44 | 20 September 2017 |
| L1C_T08VNk_A06489_20180603T | 650 | | 3 June 2018 |
| LC08_L1TP_05801920180705_T1 | 875 | 7.50 | 5 July 2018 |
| LC08_L1TP_05801920180721_T1 | 1000 | 7.81 | 21 July 2018 |
| LC08_L1TP_05701920180730_T1 | 1075 | 8.33 | 30 July 2018 |
| LC08_L1TP_05701920180916_T1 | 1400 | 6.77 | 16 September 2018 |
| LC08_L1TP_05701920181010_T1 | 1425 | 4.93 | 10 October 2018 |

**Table 2.** Taku Glacier rate of rise of the transient snow line from Landsat imagery on dates at least 15 days a part. The product of the specific migration rate for a period and the mean observed balance gradient from 900 to 1200 m.a.s.l on Taku Glacier yields the ablation rate.

| Start Date | End Date | Days | Rise | Rate of Rise | Ablation |
|---|---|---|---|---|---|
| 23 July 1984 | 18 August 1984 | 27 | 140 | 4.26 | 22 |
| 23 July 1984 | 25 August 1984 | 34 | 200 | 5.30 | 27 |
| 30 July 1998 | 16 September 1998 | 48 | 175 | 3.65 | 19 |
| 12 July 2003 | 5 August 2003 | 24 | 75 | 3.13 | 16 |
| 5 August 2003 | 22 August 2003 | 17 | 100 | 5.88 | 30 |
| 14 July 2004 | 8 August 2004 | 24 | 80 | 3.33 | 17 |
| 14 July 2004 | 16 August 2004 | 32 | 125 | 3.91 | 20 |
| 8 August 2004 | 24 August 2004 | 16 | 70 | 4.38 | 22 |
| 16 August 2004 | 31 August 2004 | 15 | 75 | 5.00 | 26 |
| 11 August 2005 | 11 September 2005 | 31 | 75 | 2.42 | 12 |
| 29 July 2006 | 15 September 2006 | 48 | 215 | 4.48 | 23 |
| 15 July 2007 | 8 August 2007 | 24 | 125 | 5.21 | 27 |
| 15 July 2007 | 16 August 2007 | 32 | 175 | 5.47 | 28 |
| 8 August 2007 | 2 September 2007 | 25 | 115 | 4.79 | 24 |
| 13 July 2009 | 5 August 2009 | 23 | 125 | 5.43 | 28 |
| 5 August 2009 | 14 September 2009 | 42 | 125 | 3.75 | 19 |
| 20 July 2011 | 12 September 2011 | 54 | 200 | 3.70 | 19 |
| 13 July 2013 | 1 August 2013 | 19 | 100 | 5.26 | 27 |
| 1 August 2013 | 2 September 2013 | 32 | 125 | 3.91 | 20 |
| 18 July 2014 | 20 August 2014 | 33 | 125 | 3.79 | 19 |
| 6 July 2015 | 14 August 2015 | 39 | 250 | 6.41 | 33 |
| 3 June 2017 | 20 September 2017 | 109 | 375 | 3.44 | 18 |
| 5 July 2018 | 21 July 2018 | 16 | 125 | 7.81 | 40 |
| 5 July 2018 | 30 July 2018 | 25 | 200 | 8.00 | 43 |
| 30 July 2018 | 16 September 2018 | 48 | 325 | 6.77 | NA |
| 30 July 2018 | 10 October 2018 | 73 | 350 | 4.93 | NA |

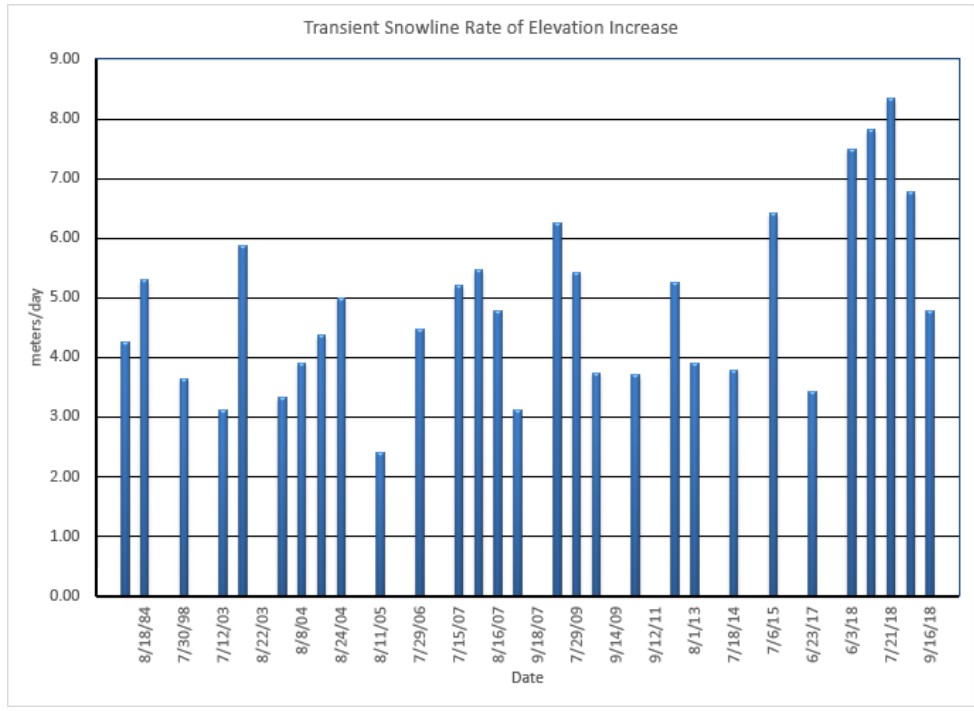

**Figure 5.** Migration rate of the TSL during specific periods of time in Table 2; the date listed is the end date of the observation period. All periods exceed 15 days.

From 10 July to 22 July 2018, eight snow pits were completed between 988 m and 1405 m on the Taku Glacier by JIRP. Retained snow water equivalent (SWE) at time of observation ranged from 0.48 m w.e. to 1.1 m w.e. Table 3. By 16 September 2018 the retained snowpack from the 2018 winter season had been lost at all eight snow pit locations. This included two locations, TK7 and MG1, that had always retained some snow cover. The ablation rate observed from the date of snow pit excavation to snow loss observed in Landsat imagery represents a minimum snowpack ablation rate. The ablation rate for specific periods from 10 July to 30 July exceeded 30 mmd$^{-1}$, 51 mmd$^{-1}$ and 58 mmd$^{-1}$. This is a larger range but encompasses the range of satellite derived ablation rate of 40 and 43 mmd$^{-1}$.

**Table 3.** Snow pit observations in 2018 on Taku Glacier by JIRP, snowpack loss was observed in ensuing satellite images.

| Snow pit | Elevation (m) | Date of Observation | SWE (m w.e.) | Complete Snowpack Loss |
|----------|---------------|---------------------|--------------|------------------------|
| TK5 | 988 | 11 July | 0.58 | 21 July |
| DG1 | 1018 | 14 July | 0.48 | 30 July |
| SW2 | 1096 | 14 July | 0.81 | 30 July |
| TK4 | 1112 | 10 July | 0.93 | 15 August |
| TK3 | 1203 | 13 July | 1.07 | 16 September |
| TK2 | 1269 | 18 July | 0.93 | 16 September |
| TK7 | 1339 | 18 July | 1.1 | 16 September |
| MG1 | 1405 | 22 July | 0.93 | 16 September |

## 4. Discussion

In 2018, the TSL and the resultant ELA was the highest observed in the 73-year annual record for Taku Glacier by 200 m. This is an exceptional expansion of the ELA range given the range of the lowest to highest observed ELA for the 1946–2017 period had been 625 m. The prior maximum of 1225 m exceeded the next highest observation by 50 m. The TSL on 21 July was near the mean ELA of the period at 1000 m, as in 2013 and 2014 (Figures 3 and 4). The TSL on 30 July was near the ELA observed in many of the recent negative balance years at 1075 m, as in 2015 and 2017 (Figures 3 and 4). The TSL migration rate observed on Taku Glacier for July 2018 and the 30 July to 16 September period is the highest observed for any period since 1984, for which repeat satellite images exist over at least a 15-day period. For two observed periods in July 2018, the TSL migration rate of 7.81 to 8.00 md$^{-1}$ yields ablation rates of 40–43 mmd$^{-1}$, approximately twice the mean ablation rate of 23 mmd$^{-1}$. This compares to observed ablation rates at snow pits ranging from 33 to 58 mmd$^{-1}$. The ablation rate is not determined for the period after 30 July as the TSL migrated above the region where the balance gradient is mapped in extensive detail.

The ablation season extends from May to September. In Juneau, Alaska, the mean monthly temperature anomaly in 2018 for May was +0.2 °C, for June was +0.3 °C, for July was +2.4 °C, for August was +0.9 °C, and for September was 0 °C [16]. The mean monthly July freezing level (1948–2018) is 2670 m for Juneau and was 3440 m in July 2018, over 200 m higher than the previous maximum, similar to the 200 m rise in maximum ELA above previous observations [17]. The exceptional July temperatures, particularly at elevation, are the key factor in driving the TSL to its highest level in the summer of 2018.

The record ELA was indicative of a large negative annual mass balance of −1.75 m, a new record minimum. The previous minimum in 1997 was −1.34 m and the mean annual balance of the 1998–2018 period was −0.22 m. On Lemon Creek Glacier, on the southwest side of the Juneau Icefield, the ELA rose above the glacier. This has occurred several other times in recent years and is not as clear an indicator of the exceptional nature of the 2018 melt season. Several hundred kilometers north in the St. Elias Range, Yukon, it was reported that rapid July ablation led to exceptional snow swamp formation on Lowell Glacier [18].

## 5. Conclusions

On Taku Glacier, the largest glacier of the Juneau Icefield, the 1946–2018 record of TSL and ELA observations indicate that 2018 had the highest ELA of the 73-year period, which at 1425 m, exceeded the previous highest observation by 200 m (1225 m in 1997). This is indicative of a strong negative balance of −1.75 m w.e. reported to the WGMS for 2018. The previous minimum mass balance was −1.34 m in 1997. The trend of increasing negative balance has been observed for Taku Glacier, Alaska and globally [4,19–21]. The rate of TSL rise during July 2018 was the highest observed to date, indicating the maximum identified ablation rate. The high ablation and TSL migration rates were driven by the warmest July on record in Juneau, AK. Will this new higher elevation location of the ELA be an anomaly or does this represent a shift in the ELA range?

**Funding:** This research received no external funding.

**Acknowledgments:** The Juneau Icefield Research Program has for more than 70 years collected annual mass balance data on the Taku Glacier. This includes the ground truth data used to derive the balance gradient and for the 2018 ablation rate validation.

**Conflicts of Interest:** The authors declare no conflict of interest.

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
