# Peer review of "Exceptionally High 2018 Equilibrium Line Altitude on Taku Glacier, Alaska"

_remotesensing, doi:10.3390/rs11202378_

Round 1

Reviewer 1 Report

The paper is about the exceptionally high equilibrium line altitude observed on Taku Glacier in 2018. While this information is interesting as a fact, it has to be put in context beyond just the findings for Taku Glacier alone and also with respect to other measures as for instance accumulation area ratio and the extensive mass balance time series available for the glacier.
Therefore, the discussion should be expanded to place the observation of one extreme event in a broader context, both geographically and methodologically. Even if data situation is not as dense as for Taku Glacier: How is the situation on the neighboring glaciers of Juneau Ice Field? Is Taku Glacier representative or an exception? In the conclusions the mass balance value for 2018 is mentioned. But how does the "strong negative balance of -1.38 m w.e." relate to previous years? Readers should not have to check WGMS themselves to place this statement into context.
The abstract contains (too) many numbers, but lacks a statement about the significance of the findings. The date format in the abstract and also throughout the text and tables should be harmonized. In my opinion, the introduction does not read fluently and needs a revision in view of sentence flow. There are several inconsistencies (e.g. in numbers between the tables, figures and text, see details below) that have to be eliminated before publication. Figure captions should be more elaborated and explain all elements shown, while maps should contain basic cartographic elements like scale, north arrow or coordinates.

Apart from these stylistic issues, there are some technical questions that have to be addressed:
1) Discuss why not all years are covered. I assume cloud cover reduces data availability? Have you tried to fill the observational gap in August 2018 (and potentially also in 2016 and 2017) with Sentinel-2 data? In the introduction you mention the availability of "many sources of satellite imagery", but you use just one.
2) The estimation of the elevation error (mean error 1.2 m) presented in section 2 does not fit to the elevation values that appear to be rounded to multiples of 5 or even 25 m. The determination of the elevation (maximum, mean of the elevation values along the TSL?) has to be explained in more detail. For a realistic elevation error evaluation spatial variability and the effect of the manual digitizing also have to be taken into account or at least discussed.

All things considered I recommend accepting the paper after moderate revision and strongly recommend the author to revise the text from a readers perspective.

Detailed comments:
l. 13 This information only appears in the abstract. Discuss this in more detail in the text. Why is the time series divided after the year 1985 into a 39-year and a 32-year period?
l. 17 The fastest rate in Table 2 is 8.0 md-1 from 7/5/2018 to 7/30/2018. Clarify.
l. 18 number of ablation rate missing?
l. 29 Glacier mass balance
l. 30 Explain to the audience of "Remote Sensing", what superimposed ice is and why maritime glacier lack it. "becomes visible" sounds like it has been buried before. I suggest to reformulate these two sentences.
l. 40 comma splice
l. 46 Check reference numbering. Furthermore, there are 4 geographic examples but 3 references, two of which refer to the same glacier.
l. 47 "from 1946-2018": has it ended? If not, since 1946.
l. 52 "comparatively consistent" What does this mean? constant in time or space? References 10, 11,13?
l. 61 Are these the same locations as the snowpits mentioned below? If not, show this transect in Figure 1.
l. 70 new paragraph for new data type?
l. 72 Are the purple dots in Figures 3 and 4 the manually digitized points? What is the distance between the points? Explain why this cannot be done automatically.
l. 78 TSL usually does not coincide with isolines and thus has covered a elevation range. State how you derive one elevation value.
l. 81 Discuss variability for different parts of the glacier. 41 out of 44 TSL values in Table 1 are rounded to multiples of 25 m, the rest are multiples of 5 m. The error "of ± 1-2 m in TSL elevation, with a mean of 1.2 m" really applies to these values?
Figure 1 There is neither scale nor a north arrow in the main map and also coordinates are missing. The caption has to be more informative. State date of the satellite image and mention and label (a,b) the insets. I suggest to show the surrounding of the glacier and the glacier outline instead of clipping the glacier. The arrow on upper map should be better discernable.
l. 91-92 Inconsistent sentence.
Figure 2: Delete colored background
l. 106 Replace "The July 30, 2018 images" with "The TSL in the July 30, 2018 image"
l. 108 In Fig. 5 there are 5 bars larger than 6.7 md-1
l. 109 "The lower rate": lower than what?
Table 1 and 2 Use same date format, preferable the one of Table 1. Numbers in Tab. 1 and Tab. 2 are inconsistent, eg. dates, rise (=TSL difference) and rise rates for 1984 are different in Tab. 2 from the numbers in Tab. 1. I have not checked the rest of the tables.
Check if Table 1 is necessary; it seems redundant to me or could be places in supplementary material.
Table 2 Give units. Column Gradient is redundant, as all rows are the same, except the two last ones being NaN which is commented in the text. Check again numbers as ablation rate is NOT the product of rate of rise and gradient in th first 2 lines.
Figure 3 Give more information in caption or legend: big and small purple dots, black dots? I suggest swapping 7/30 and 7/21, so the sequence is upper panel left to right and then lower panel.

Figure 5 There are additional bars plotted, that are not listed in Tab. 1 nor Tab. 2, e.g. the 1993 one. Where do the 8 additional data points come from?
l. 136 delete "?"
l. 140 put TK7 and MG1 in parentheses or behind "locations"
l. 142 Check superscript in units (and spaces). Refer to Table 3. The expression "various periods" is strange here. From Table 3 there are 3 observations that give these numbers, but they refer to 2 periods (10-Jul to 21-Jul and 14-Jul to 30-Jul). And they refer to 3 places, so there is variation in space and time.
l. 144 This is Table 3 (unless you delete Table 1)
l. 152 refer to Figure 3 and/or 4 but not 5.
l. 154 Use plural: TSL migration rates .. are.
l. 163 Is reference 17 correct here?
l. 168 That the highest ELA exceeds the previous highest is obvious. Maybe: "exceeded the previous highest observation recorded in 1998 by 200 m". Compare the 2018 mass balance to previous years. This should be expanded in the discussion section.
l. 185 something missing?
l. 198 doi:10.3189/2013AoG62A083
Check punctuation in the references and harmonize the style of doi numbers.

Author Response

REV 1
The paper is about the exceptionally high equilibrium line altitude observed on Taku Glacier in 2018. While this information is interesting as a fact, it has to be put in context beyond just the findings for Taku Glacier alone and also with respect to other measures as for instance accumulation area ratio and the extensive mass balance time series available for the glacier.

Response:The ELA is used to define Accumulation area ratio, separate reporting does not add to this.  Mass balance information has been added. The goal is keep focus on the TSL observations, mass balance is based on field measurements.

Therefore, the discussion should be expanded to place the observation of one extreme event in a broader context, both geographically and methodologically. Even if data situation is not as dense as for Taku Glacier: How is the situation on the neighboring glaciers of Juneau Ice Field? Is Taku Glacier representative or an exception? In the conclusions the mass balance value for 2018 is mentioned. But how does the "strong negative balance of -1.38 m w.e." relate to previous years? Readers should not have to check WGMS themselves to place this statement into context.

Response: This is recent data and has not been published elsewhere yet except at WGMS.  I have added data on Lemon Creek Glacier which is also reported at WGMS. Also added reference to Lowell Glacier in an article from NASA Earth Observatory and an unusual snow swamp. “On Lemon Creek Glacier on the southwest side of the Juneau Icefield the ELA rose above the glacier.  This has occurred several other times in recent years and is not as clear an indicator or the exceptional nature of the 2018 melt season. Several hundred kilometers north in the St. Elias Range, Yukon it was reported in 2018 that rapid July ablation led to an exceptional snow swamp formation on Lowell Glacier [18]. “

The abstract contains (too) many numbers, but lacks a statement about the significance of the findings. The date format in the abstract and also throughout the text and tables should be harmonized. In my opinion, the introduction does not read fluently and needs a revision in view of sentence flow. There are several inconsistencies (e.g. in numbers between the tables, figures and text, see details below) that have to be eliminated before publication. Figure captions should be more elaborated and explain all elements shown, while maps should contain basic cartographic elements like scale, north arrow or coordinates. Each figure and or caption has been adjusted.
Apart from these stylistic issues, there are some technical questions that have to be addressed:
1) Discuss why not all years are covered. I assume cloud cover reduces data availability? Have you tried to fill the observational gap in August 2018 (and potentially also in 2016 and 2017) with Sentinel-2 data? In the introduction you mention the availability of "many sources of satellite imagery", but you use just one.

Response: I do use Sentinel as well, but cloud cover prevents usable imagery for this analysis in 2016, 2017 or 2018. There is a limited elevation range that is the focus and the focus is on the period of rising snowline. There are usable single images to identify the ELA in 2016 and 2017.

2) The estimation of the elevation error (mean error 1.2 m) presented in section 2 does not fit to the elevation values that appear to be rounded to multiples of 5 or even 25 m. The determination of the elevation (maximum, mean of the elevation values along the TSL?) has to be explained in more detail. For a realistic elevation error evaluation spatial variability and the effect of the manual digitizing also have to be taken into account or at least discussed.

Response: We report the ELA or TSL to the nearest 5 m as it is an average elevation of a number of points on the glacier surface. 

All things considered I recommend accepting the paper after moderate revision and strongly recommend the author to revise the text from a readers perspective.
Detailed comments:
l. 13 This information only appears in the abstract. Discuss this in more detail in the text. Why is the time series divided after the year 1985 into a 39-year and a 32-year period? Text is added.
l. 17 The fastest rate in Table 2 is 8.0 md-1 from 7/5/2018 to 7/30/2018. Clarify. 
Changed to this number and period.

l. 18 number of ablation rate missing?-added
l. 29 Glacier mass balance- mass added
l. 30 Explain to the audience of "Remote Sensing", what superimposed ice is and why maritime glacier lack it. "becomes visible" sounds like it has been buried before. I suggest to reformulate these two sentences. 
Response: Superimposed ice develops when percolating meltwater refreezes on a cold underlying glacier surface during summer, on temperate glaciers a cold enough surface does not exist in summer.

l. 46 Check reference numbering. Furthermore, there are 4 geographic examples but 3 references, two of which refer to the same glacier.-
Reference added.

l. 47 "from 1946-2018": has it ended? If not, since 1946.
-Ongoing added.
l. 52 "comparatively consistent" What does this mean? constant in time or space? References 10, 11,13? -Changed and corrected reference
l. 61 Are these the same locations as the snowpits mentioned below? If not, show this transect in Figure 1. -The transect is not the same. It is now shown.
l. 70 new paragraph for new data type? -Okay
l. 72 Are the purple dots in Figures 3 and 4 the manually digitized points? What is the distance between the points? Explain why this cannot be done automatically. 
response: The TSL could be identified automatically, but it is more accurate when examining a single glacier to do so manually.  The dots do not have a constant spacing, instead they are spaced so as not to visually obscure the snowline for the reader.
l. 78 TSL usually does not coincide with isolines and thus has covered a elevation range. State how you derive one elevation value. 
Response: It is the mean observed elevation on the profile across the main branch of the Taku Glacier.
l. 81 Discuss variability for different parts of the glacier. 41 out of 44 TSL values in Table 1 are rounded to multiples of 25 m, the rest are multiples of 5 m. The error "of ± 1-2 m in TSL elevation, with a mean of 1.2 m" really applies to these values? 
Sentence added, We report the ELA or TSL to the nearest 5 m as it is an average elevation of a number of points on the glacier surface. 

Figure 1 There is neither scale nor a north arrow in the main map and also coordinates are missing. The caption has to be more informative. State date of the satellite image and mention and label (a,b) the insets. I suggest to show the surrounding of the glacier and the glacier outline instead of clipping the glacier. The arrow on upper map should be better discernable.
Response: The image has an added scale and North arrow.  The clipped image could be changed, but I think it focusses the reader, the visual design is specific to bring out the location of snowpits vis a vis the snowline and the other two reviewers saw no issue.

l. 91-92 Inconsistent sentence.- Adjusted
Figure 2: Delete colored background. 
- I can delete colored background, what is the preference of the editors? I think visually it is good.
l. 106 Replace "The July 30, 2018 images" with "The TSL in the July 30, 2018 image"  -Corrected
l. 108 In Fig. 5 there are 5 bars larger than 6.7 md-1.0-  Corrected in text
l. 109 "The lower rate": lower than what? -Added text

Table 1 and 2 Use same date format, preferable the one of Table 1. Numbers in Tab. 1 and Tab. 2 are inconsistent, eg. dates, rise (=TSL difference) and rise rates for 1984 are different in Tab. 2 from the numbers in Tab. 1. I have not checked the rest of the tables. 
-Yes rows switched in 1984 corrected. 

Check if Table 1 is necessary; it seems redundant to me or could be places in supplementary material.
Response: Reviewer 2 indicates it is
Table 2 Give units. Column Gradient is redundant, as all rows are the same, except the two last ones being NaN which is commented in the text. Check again numbers as ablation rate is NOT the product of rate of rise and gradient in the first 2 lines. 
Response: Gradient removed. The first two rows values reversed, now corrected.

Figure 3 Give more information in caption or legend: big and small purple dots, black dots? I suggest swapping 7/30 and 7/21, so the sequence is upper panel left to right and then lower panel. The comparison of start to end of July is important and that is why they are adjacent in top panel, it is not a random choice.-Caption expanded.
Figure 5 There are additional bars plotted, that are not listed in Tab. 1 nor Tab. 2, e.g. the 1993 one. Where do the 8 additional data points come from? -
A new figure with correct dates created
l. 136 delete "?"- removed
l. 140 put TK7 and MG1 in parentheses or behind "locations"- moved
l. 142 Check superscript in units (and spaces). Refer to Table 3. The expression "various periods" is strange here. From Table 3 there are 3 observations that give these numbers, but they refer to 2 periods.  (10-Jul to 21-Jul and 14-Jul to 30-Jul). And they refer to 3 places, so there is variation in space and time.- Changed to specific
l. 144 This is Table 3 (unless you delete Table 1)- corrected
l. 152 refer to Figure 3 and/or 4 but not 5.- changed
l. 154 Use plural: TSL migration rates .. are. -changed
l. 163 Is reference 17 correct here? -Is not used and should not be
l. 168 That the highest ELA exceeds the previous highest is obvious. Maybe: "exceeded the previous highest observation recorded in 1998 by 200 m". Compare the 2018 mass balance to previous years. This should be expanded in the discussion section. 
Response:  The record ELA was indicative of a large negative annual mass balance of -1.75 m a new record minimum.  The previous minimum in 1997 was -1.34 m and the mean annual balance of the 1998-2018 period was -0.22 m.  
l. 185 something missing? -added
l. 198 doi:10.3189/2013AoG62A083 -Okay.

Reviewer 2 Report

Comments for Remote Sensing by Dr Ian S. Evans, Durham University, on –

Exceptionally High 2018 Equilibrium Line Altitude on Taku Glacier, Alaska – by  M Pelto

                More bad news about glacier loss – thanks!  The length of observation period makes the outlier nature of the 2018 melt even more impressive.  The illustrations are good (but the black and purple dots need defining) and the writing just needs some tidying up, as noted below.  The data are well-presented, and interpreted carefully and cautiously.  It is interesting to see TSL used to calculate ablation rate, and produce results consistent with snowpit data.  (Is the ‘NA’ ablation in the last two rows of Table 2 because TSL has left the 900 – 1200 m range for which ablation gradient is available?  It might be worth mentioning… .)  The link to high temperatures in Juneau is clear.

                Have you correlated rate of TSL rise with length of period, with elevation (middle of the two), and with Julian date?  That is worth checking.

                Figure 2 does invite rather more discussion (even if these results have been discussed elsewhere).  My take on the trends would be: declining ELA from 1946 to 1972, and increasing from 1986 on, especially from 2001 on.  Alternatively, oscillation between 1988 and 2013, between the declining and increasing trends.  It is interesting that the ELA rise is rather late, only the 4 years from 2015 on have been consistently high.  Compared with accelerating losses throughout the world, most have been high since 2000.  Would it be worth speculating why Taku has been delayed in catching up?

                I thought about proposing combining Tables 1 and 2, as they have two columns (rate and end-date) in common, but decided it is clearer to keep them as they are.

                It would be worth emphasising that each TSL-derived ablation rate is specific to the altitude range over which TSL moved (cf. lines  120, 126, 156 and 171).

                With annual ELAs so far above balance ELA0s, perhaps we should talk about ‘disequilibrium lines’!  [only joking…]

DETAILED SUGGESTIONS:  lines -

17-18, 92, 107 & 109  I think there should be a space between m and d (here and elsewhere) – as these are not ‘metre-days’.  Likewise mmd (lines 119-121, 142-143 & 157-158).

18 Number is missing, before md-1

19 ’16 September’ rather than ‘9/16’ would be more consistent with the rest of the abstract.

43 delete second ‘the’

53-54 How can “the lower ablation zone  … that the TSL does not transect” be true?  Snowfall extends down to sea level, at least in some years (even as far south as Vancouver!).   Presumably there is some other reason why TSL was not measured in the lower ablation zone – so it would be a good idea to explain why:  because TSL at start of field season has already risen above lowest part of ablation zone?

56-58  It would be good to amplify this presentation of two averages.  There were variations within each of the two periods.  Discussion of the trend (possibly curvilinear) or trends would be helpful.  (Also, looking at it simplistically, the 7.83 m loss since 1988 comes nowhere near disposing of the 17.64 m surplus from 1946-88, so the glacier is still much bigger than in 1946 – is that true?)

57-58 I think there should be a space between m and a.

66 ‘using a’

90 ‘made, at’

92 delete ‘this’ , or change the sentence.

99 Insert  (month/day/year) after first use of this strangely illogical American convention …

106 ‘image gives approximately …’

107 ‘rate was’

108 ‘there were’

119 comma after ‘periods’, or re-write sentence.

132 Presumably purple dots are max. TSL: what are the black dots?  Both should be defined in caption.  Why not use lines, or less coarse dots?

136 delete ‘?’  (and ‘the’?)

140 comma after ‘snowcover’?

142-143 & 158  All ‘-1’ must be subscripted

151 & 152 ‘was’ rather than ‘is’

152 perhaps ‘as in’ rather than ‘note’

153 perhaps ‘e.g.’ rather than ‘note’

References: there are various format problems, e.g. sometimes a space between volume and issue number, sometimes not.  Some volume numbers are italicised, others not.

183 delete last comma

185 complete page numbers

202 Insert page numbers (or length)

204 Issue number is not bracketed (as elsewhere)

215 Insert page numbers (or length)

217 insert year

177 is irrelevant -delete

178 ‘author’

Author Response

Rev 2
    More bad news about glacier loss – thanks!  The length of observation period makes the outlier nature of the 2018 melt even more impressive.  The illustrations are good (but the black and purple dots need defining) and the writing just needs some tidying up, as noted below.  The data are well-presented, and interpreted carefully and cautiously.  It is interesting to see TSL used to calculate ablation rate, and produce results consistent with snowpit data.  (Is the ‘NA’ ablation in the last two rows of Table 2 because TSL has left the 900 – 1200 m range for which ablation gradient is available?  It might be worth mentioning… .)  The link to high temperatures in Juneau is clear.  

Response:Sentence added to explain. The ablation rate is not determined for the period after July 30 as the TSL migrated above the region where the balance gradient is mapped in extensive detail.
                Have you correlated rate of TSL rise with length of period, with elevation (middle of the two), and with Julian date?  That is worth checking. TSL is reported as daily rate so correlation with length of period makes little sense. 

Response: Until 2018 there is a narrow range of elevation that is impacted and there is not a correlation with elevation, with just one year in the range above 1200 m, not much can be said about the rate.
                Figure 2 does invite rather more discussion (even if these results have been discussed elsewhere).  My take on the trends would be: declining ELA from 1946 to 1972, and increasing from 1986 on, especially from 2001 on.  Alternatively, oscillation between 1988 and 2013, between the declining and increasing trends.  It is interesting that the ELA rise is rather late, only the 4 years from 2015 on have been consistently high.  Compared with accelerating losses throughout the world, most have been high since 2000.  Would it be worth speculating why Taku has been delayed in catching up? This is not an accurate description of what has happened on Taku Glacier, which has followed global trends, the last four years have been exceptional globally too. 
                I thought about proposing combining Tables 1 and 2, as they have two columns (rate and end-date) in common, but decided it is clearer to keep them as they are.  I will keep both
                It would be worth emphasising that each TSL-derived ablation rate is specific to the altitude range over which TSL moved (cf. lines  120, 126, 156 and 171). -
-This sentence is added
                With annual ELAs so far above balance ELA0s, perhaps we should talk about ‘disequilibrium lines’!  [only joking…] true
DETAILED SUGGESTIONS:  lines -
17-18, 92, 107 & 109  I think there should be a space between m and d (here and elsewhere) – as these are not ‘metre-days’.  Likewise mmd (lines 119-121, 142-143 & 157-158).
Response: I disagree but will do so if RS editors think there should be.
18 Number is missing, before md-1-
added
19 ’16 September’ rather than ‘9/16’ would be more consistent with the rest of the abstract. Date changed. 
-Date changed
43 delete second ‘the’
r-emoved
53-54 How can “the lower ablation zone  … that the TSL does not transect” be true?  Snowfall extends down to sea level, at least in some years (even as far south as Vancouver!).   Presumably there is some other reason why TSL was not measured in the lower ablation zone – so it would be a good idea to explain why:  because TSL at start of field season has already risen above lowest part of ablation zone?
Corrected- where the balance gradient is not measured annually or in the upper accumulation zone that the TSL does not transect.
56-58  It would be good to amplify this presentation of two averages.  There were variations within each of the two periods.  Discussion of the trend (possibly curvilinear) or trends would be helpful.  (Also, looking at it simplistically, the 7.83 m loss since 1988 comes nowhere near disposing of the 17.64 m surplus from 1946-88, so the glacier is still much bigger than in 1946 – is that true?)
Response: Period changed to match ELA there is not a curvilinear trend, nor is mass balance or trend analysis the focus of this paper.
57-58 I think there should be a space between m and a.
I disagree but will do so if RS editors think there should be.
66 ‘using a’-
added
90 ‘made, at’-
added
92 delete ‘this’ , or change the sentence.
Removed-
99 Insert  (month/day/year) after first use of this strangely illogical American convention …
added
106 ‘image gives approximately …’ -corrected
107 ‘rate was’ -corrected
108 ‘there were’ -corrected
132 Presumably purple dots are max. TSL: what are the black dots?  Both should be defined in caption.  Why not use lines, or less coarse dots?

Response: Lines would visually obscure the actual snowline.  Smaller dots could be used, but visually given the size of figures this dots size is appropriate for best visual display. Caption fixed.

136 delete ‘?’  (and ‘the’?)- corrected
140 comma after ‘snowcover’? -corrected
142-143 & 158  All ‘-1’ must be subscripted- corrected
151 & 152 ‘was’ rather than ‘is’ -corrected
152 perhaps ‘as in’ rather than ‘note’- corrected
153 perhaps ‘e.g.’ rather than ‘note’- corrected
References: there are various format problems, e.g. sometimes a space between volume and issue number, sometimes not.  Some volume numbers are italicised, others not.
corrected
183 delete last comma-- corrected
185 complete page numbers -corrected
202 Insert page numbers (or length)- corrected
215 Insert page numbers (or length)- added
217 insert year-corrected
177 is irrelevant -delete  corrected
178 ‘author’

Reviewer 3 Report

The author’ goal seems to report the ELA variation from 1984 to 2018, especially in 2018 by using satellite data. This work is significant for surveying glacier melting. This manuscript is generally well-written. Thus, I have to suggest that it could be released.

Table 2 8/25/1984? Is it 1985?

Line 136 From July 10 to July 22,2018? Please check the date.

Line 159 please give the sources of temperature

Author Response

The author’ goal seems to report the ELA variation from 1984 to 2018, especially in 2018 by using satellite data. This work is significant for surveying glacier melting. This manuscript is generally well-written. Thus, I have to suggest that it could be released.

Table 2 8/25/1984? Is it 1985? -No
Line 136 From July 10 to July 22,2018? Please check the date.- It is correct
Line 159 please give the sources of temperature- Given now